# Automatic Goal Generation for Reinforcement Learning Agents

## Abstract

Reinforcement learning (RL) is a powerful technique to train an agent to perform a task. However, an agent that is trained using RL is only capable of achieving the single task that is specified via its reward function. Such an approach does not scale well to settings in which an agent needs to perform a diverse set of tasks, such as navigating to varying positions in a room or moving objects to varying locations. Instead, we propose a method that allows an agent to automatically discover the range of tasks that it is capable of performing in its environment. We use a generator network to propose tasks for the agent to try to accomplish, each task being specified as reaching a certain parametrized subset of the state-space. The generator network is optimized using adversarial training to produce tasks that are always at the appropriate level of difficulty for the agent. Our method thus automatically produces a curriculum of tasks for the agent to learn. We show that, by using this framework, an agent can efficiently and automatically learn to perform a wide set of tasks without requiring any prior knowledge of its environment[1]. Our method can also learn to accomplish tasks with sparse rewards, which pose significant challenges for traditional RL methods.

## 1 Introduction

Reinforcement learning (RL) can be used to train an agent to perform a task by optimizing a reward function. Recently, a number of impressive results have been demonstrated by training agents using RL: such agents have been trained to defeat a champion Go player (Silver et al., 2016), to outperform humans in 49 Atari games (Guo et al., 2016; Mnih et al., 2015), and to perform a variety of difficult robotics tasks (Lillicrap et al., 2015; Duan et al., 2016; Levine et al., 2016). In each of the above cases, the agent is trained to optimize a single reward function in order to learn to perform a single task. However, there are many real-world environments in which a robot will need to be able to perform not a single task but a diverse set of tasks, such as navigating to varying positions in a room or moving objects to varying locations. We consider the problem of maximizing the average success rate of our agent over all possible goals, where success is defined as the probability of successfully reaching each goal by the current policy.

In order to efficiently maximize this objective, the algorithm must intelligently choose which goals to focus on at every training stage: goals should be at the appropriate level of difficulty for the current policy. To do so, our algorithm allows an agent to generate its own reward functions, defined with respect to target subsets of the state space, called goals. We generate such goals using a Goal Generative Adversarial Network (Goal GAN), a variation of to the GANs introduced by Goodfellow et al. (2014). A goal discriminator is trained to evaluate whether a goal is at the appropriate level of difficulty for the current policy, and a goal generator is trained to generate goals that meet this criteria. We show that such a framework allows an agent to quickly learn a policy that reaches all feasible goals in its environment, with no prior knowledge about the environment or the tasks being performed. Our method automatically creates a curriculum, in which, at each step, the generator generates goals that are only slightly more difficult than the goals that the agent already knows how to achieve.

In summary, our main contribution is a method for automatic curriculum generation that considerably improves the sample efficiency of learning to reach all feasible goals in the environment.

---

[1]Videos and code available at: `https://sites.google.com/view/goalgeneration4rl`

Learning to reach multiple goals is useful for multi-task settings such as navigation or manipulation, in which we want the agent to perform a wide range of tasks. Our method also naturally handles sparse reward functions, without needing to manually modify the reward function for every task, based on prior task knowledge. Instead, our method dynamically modifies the probability distribution from which goals are sampled to ensure that the generated goals are always at the appropriate difficulty level, until the agent learns to reach all goals within the feasible goal space.

## 2 RELATED WORK

The problem that we are exploring has been referred to as "multi-task policy search" (Deisenroth et al., 2014) or "contextual policy search," in which the task is viewed as the context for the policy (Deisenroth et al., 2013; Fabisch & Metzen, 2014). Unlike the work of Deisenroth et al. (2014), our work uses a curriculum to perform efficient multi-task learning, even in sparse reward settings. In contrast to Fabisch & Metzen (2014), which trains from a small number of discrete contexts / tasks, our method generates a training curriculum directly in continuous task space.

**Intrinsic Motivation:** Intrinsic motivation involves learning with an intrinsically specified objective (Schmidhuber, 1991; 2010). Intrinsic motivation has also been studied extensively in the developmental robotics community, such as SAGG-RIAC (Baranes & Oudeyer, 2010; 2013a), which has a similar goal of learning to explore a parameterized task space. However, our experiments with SAGG-RIAC demonstrate that this approach do not explore the space as efficiently as ours. A related concept is that of competence-based intrinsic motivation (Baldassarre & Mirolli, 2012), which uses a selector to select from a discrete set of experts. Recently there have been other formulations of intrinsic motivation, relating to optimizing surprise (Houthooft et al., 2016; Achiam & Sastry, 2016) or surrogates of state-visitation counts (Bellemare et al., 2016; Tang et al., 2016). All these approaches improve learning in sparse tasks where naive exploration performs poorly. However, these formulations do not have a notion of which states are hard for the learner, and the intrinsic motivation is independent of the current performance of the agent. In contrast, our formulation of intrinsic motivation directly relates to our policy performance: the agent is motivated to train on tasks that push the boundaries of its capabilities.

**Curriculum Learning:** The increasing interest on training single agents to perform multiple tasks is leading to new developments on how to optimally present the tasks to the agent during learning. The idea of using a curriculum has been explored in many prior works on supervised learning (Bengio et al., 2009; Zaremba & Sutskever, 2014; Bengio et al., 2015). However, these curricula are usually hand-designed, using the expertise of the system designer. Another lines of work take into explicit consideration which examples are hard for the current learner (Kumar et al., 2010; Jiang et al., 2015); or use learning progress to build an automatic curriculum(Graves et al., 2017), however both approaches have mainly been applied for supervised tasks. Most curriculum learning in RL still relies on fixed pre-specified sequences of tasks (Karpathy & Van De Panne, 2012). Other recent work has proposed using a given baseline performance for several tasks to gauge which tasks are the hardest and require more training (Sharma & Ravindran, 2017), but the framework can only handle a finite set of tasks and cannot handle sparse rewards. Our method trains a policy that generalizes to a set of continuously parameterized tasks and is shown to perform well even under sparse rewards by not allocating training effort to tasks that are too hard for the current performance of the agent. Finally, an interesting self-play strategy has been proposed that is concurrent to our work (Sukhbaatar et al., 2017); however, they view their approach as simply providing an exploration bonus for a single target task; in contrast, we focus on the problem of efficiently optimizing a policy across a range of goals, as we explain below.

## 3 PROBLEM DEFINITION

### 3.1 GOAL-PARAMETERIZED REWARD FUNCTIONS

In the traditional RL framework, at each timestep $t$, the agent in state $s_t \in \mathcal{S} \subseteq \mathbb{R}^n$ takes an action $a_t \in \mathcal{A} \subseteq \mathbb{R}^m$, according to some policy $\pi(a_t|s_t)$ that maps from the current state $s_t$ to a probability distribution over actions. Taking this action causes the agent to enter into a new state $s_{t+1}$ according to a transition distribution $p(s_{t+1}|s_t, a_t)$, and receive a reward $r_t = r(s_t, a_t, s_{t+1})$. The objective

of the agent is to find the policy $\pi$ that maximizes the expected return, defined as the sum of rewards $R = \sum_{t=0}^{T} r_t$, where $T$ is a maximal time given to perform the task. The learned policy corresponds to maximizing the expected return for a single reward function.

In our framework, instead of learning to optimize a single reward function, we consider a range of reward functions $r^g$ indexed or parametrized by a goal $g \in \mathcal{G}$. Each goal $g$ corresponds to a set of states $S^g \subset \mathcal{S}$ such that goal $g$ is considered to be achieved when the agent is in any state $s_t \in S^g$. Then the objective is to learn a policy that, given any goal $g \in \mathcal{G}$, acts optimally with respect to $r^g$. We define a very simple reward function that measures whether the agent has reached the goal:

$$r^g(s_t, a_t, s_{t+1}) = \mathbb{1}\{s_{t+1} \in S^g\}, \tag{1}$$

where $\mathbb{1}$ is the indicator function. In our case, we use $S^g = \{s_t : d(f(s_t), g) \leq \epsilon\}$, where $f(\cdot)$ is a function that projects a state into goal space $\mathcal{G}$, $d(\cdot, \cdot)$ is a distance metric in goal space, and $\epsilon$ is the acceptable tolerance that determines when the goal is reached. However, our method can handle generic binary rewards (as in Eq. (1)) and does not require a distance metric for learning. Furthermore, we define our MDP such that each episode terminates when $s_t \in S^g$. Thus, the return $R^g = \sum_{t=0}^{T} r_t^g$ is a binary random variable whose value indicates whether the agent has reached the set $S^g$ in at most $T$ time-steps. Hence, the return of a trajectory $s_0, s_1, \ldots$ can be expressed as $R^g = \mathbb{1}\{\bigcup_{t=0}^{T} s_t \in S^g\}$. Now, policies are also conditioned on the current goal $g$ (as in Schaul et al. (2015)), written as $\pi(a_t \mid s_t, g)$. The expected return obtained when we take actions sampled from the policy can then be expressed as the probability of succeeding on each goal within T timesteps, as shown in Eq. (2).

$$R^g(\pi) = \mathbb{E}_{\pi(\cdot \mid s_t, g)} \mathbb{1}\{\exists\, t \in [1 \ldots T] : s_t \in S^g\} = \mathbb{P}\left(\exists\, t \in [1 \ldots T] : s_t \in S^g \,\Big|\, \pi, g\right) \tag{2}$$

The sparse indicator reward function of Eq. (1) is not only simple but also represents a property of many real-world goal problems: in many settings, it may be difficult to tell whether the agent is getting closer to achieving a goal, but easy to tell when a goal has been achieved. For example, for a robot moving in a maze, taking actions that maximally reduce the straight-line distance from the start to the goal is usually not a feasible approach for reaching the goal, due to the presence of obstacles along the path. In theory, one could hand-engineer a meaningful distance function for each task that could be used to create a dense reward function. Instead, we use the indicator function of Eq. (1), which simply captures our objective by measuring whether the agent has reached the goal state. We show that our method is able to learn even with such sparse rewards.

## 3.2 OVERALL OBJECTIVE

We desire to find a policy $\pi(a_t \mid s_t, g)$ that achieves a high reward for many goals $g$. We assume that there is a test distribution of goals $p_g(g)$ that we would like to perform well on. For simplicity, we assume that the test distribution samples goals uniformly from the set of goals $\mathcal{G}$, although in practice any distribution can be used. The overall objective is then to find a policy $\pi^*$ such that

$$\pi^*(a_t \mid s_t, g) = \arg\max_{\pi} \mathbb{E}_{g \sim p_g(\cdot)} R^g(\pi). \tag{3}$$

Recall from Eq. (2) that $R^g(\pi)$ is the probability of success for each goal $g$. Thus the objective of Eq. (3) measures the average probability of success over all goals sampled from $p_g(g)$. We refer to the objective in Eq. (3) as the coverage objective.

## 3.3 ASSUMPTIONS

Similar to previous work (Schaul et al., 2015; Kupcsik et al., 2013; Fabisch & Metzen, 2014; Deisenroth et al., 2014) we need a continuous goal-space representation such that a goal-conditioned policy can efficiently generalize over the goals. In particular, we assume that:

1. A policy trained on a sufficient number of goals in some area of the goal-space will learn to interpolate to other goals within that area.
2. A policy trained on some set of goals will provide a good initialization for learning to extrapolate to close-by goals, meaning that the policy can occasionally reach them but maybe not consistently so.

Furthermore, we assume that if a goal is reachable, there exists a policy that does so reliably. This is a reasonable assumption for any practical robotics problem, and it will be key for our method, as it strives to train on every goal until it is consistently reached.

## 4 METHOD

Our approach can be broken down into three parts: First, we label a set of goals based on whether they are at the appropriate level of difficulty for the current policy. Second, using these labeled goals, we construct and train a generator to output new goals that are at the appropriate level of difficulty. Finally, we use these new goals to efficiently train the policy, improving its coverage objective. We iterate through each of these steps until the policy converges.

### 4.1 GOAL LABELING

As shown in our experiments, sampling goals from $p_g(g)$ directly, and training our policy on each sampled goal may not be the most sample efficient way to optimize the coverage objective of Eq. (3). Instead, we modify the distribution from which we sample goals during training: we wish to find the set of goals $g$ in the set $G_i = \{g : R_{\min} \leq R^g(\pi_i) \leq R_{\max}\} \subseteq \mathcal{G}$.

The justification for this is as follows: due to the sparsity of the reward function, for most goals $g$, the current policy $\pi_i$ (at iteration $i$) obtains no reward. Instead, we wish to train our policy on goals $g$ for which $\pi_i$ is able to receive some minimum expected return $R^g(\pi_i) > R_{\min}$ such that the agent receives enough reward signal for learning. On the other hand, if we only sample from goals for which $R^g(\pi_i) > R_{\min}$, we might sample repeatedly from a small set of already mastered goals. To force our policy to train on goals that still need improvement, we train on the set of goals $g$ for which $R^g(\pi_i) \leq R_{\max}$, where $R_{\max}$ is a hyperparameter setting a maximum level of performance above which we prefer to concentrate on new goals. Thus, training our policy on goals in $G_i$ allows us to efficiently maximize the coverage objective of Eq. (3). Note that from Eq. (2), $R_{\min}$ and $R_{\max}$ can be interpreted as a minimum and maximum probability of reaching a goal over $T$ timesteps. Given a set of goals sampled from some distribution $p_{\text{data}}(g)$, we wish to estimate a label $y_g \in \{0, 1\}$ for each goal $g$ that indicates whether $g \in G_i$. These labels are obtained based on the policy performance during the policy update step (Sec. 4.3); see Appendix C for details on this procedure. In the next section we describe how we can generate more goals that also belong to $G_i$, in addition to the goals that we have labeled.

### 4.2 ADVERSARIAL GOAL GENERATION

In order to sample new goals $g$ uniformly from $G_i$, we introduce an adversarial training procedure called "goal GAN", which is a modification of the procedure used for training Generative Adversarial Networks (GANs) (Goodfellow et al., 2014). The modification allows us to train the generative model both with positive examples from the distribution we want to approximate and negative examples sampled from a distribution that does not share support with the desired one. This improves the accuracy of the generative model despite being trained with very few positive samples. Our choice of GANs for goal generation was motivated both from this potential to train from negative examples as well as their ability to generate very high dimensional samples such as images (Goodfellow et al., 2014) which is important for scaling up our approach to goal generation in high-dimensional goal spaces. Other generative models like Stochastic Neural Networks (Tang & Salakhutdinov, 2013) don't accept negative examples and don't have the potential to scale up to higher dimensions.

In our particular application, we use a "goal generator" neural network $G(z)$ to generate goals $g$ from a noise vector $z$. We train the goal generator to uniformly output goals in $G_i$ using a second "goal discriminator" network $D(g)$. The latter is trained to distinguish goals that are in $G_i$ from goals that are not in $G_i$. We optimize our $G(z)$ and $D(g)$ in a manner similar to that of the Least-Squares GAN (LSGAN) (Mao et al., 2016), which we modify by introducing the binary label $y_g$ to indicate whether $g \in G_i$ (allowing us to train from "negative examples" when $y_g = 0$):

$$\min_D V(D) = \mathbb{E}_{g \sim p_{\text{data}}(g)} \left[ y_g (D(g) - b)^2 + (1 - y_g)(D(g) - a)^2 \right] + \mathbb{E}_{z \sim p_z(z)}[(D(G(z)) - a)^2]$$

$$\min_G V(G) = \mathbb{E}_{z \sim p_z(z)}[D(G(z)) - c)^2] \tag{4}$$

We directly use the original hyperparameters reported in Mao et al. (2016) in all our experiments (a = -1, b = 1, and c = 0). The LSGAN approach gives us a considerable improvement in training stability over vanilla GAN, and it has a comparable performance to WGAN (Arjovsky et al., 2017). However, unlike in the original LSGAN paper (Mao et al., 2016), we have three terms in our value function $V(D)$ rather than the original two. For goals $g$ for which $y_g = 1$, the second term disappears and we are left with only the first and third terms, which are identical to that of the original LSGAN framework. Viewed in this manner, the discriminator is trained to discriminate between goals from $p_{\text{data}}(g)$ with a label $y_g = 1$ and the generated goals $G(z)$. Looking at the second term, our discriminator is also trained with "negative examples," i.e. goals with a label $y_g = 0$ which our generator should not generate. The generator is trained to "fool" the discriminator, i.e. to output goals that match the distribution of goals in $p_{\text{data}}(g)$ for which $y_g = 1$.

## 4.3 POLICY OPTIMIZATION

---

**Algorithm 1:** Generative Goal Learning

**Input** : Policy $\pi_0$
**Output:** Policy $\pi_N$
$(G, D) \leftarrow \texttt{initialize\_GAN}()$
$goals_{\text{old}} \leftarrow \varnothing$
**for** $i \leftarrow 1$ **to** $N$ **do**
   $z \leftarrow \texttt{sample\_noise}(p_z(\cdot));$
   $goals \leftarrow G(z) \cup \texttt{sample}(goals_{\text{old}});$
   $\pi_i \leftarrow \texttt{update\_policy}(goals, \pi_{i-1});$
   $returns \leftarrow$
    $\texttt{evaluate\_policy}(goals, \pi_i);$
   $labels \leftarrow \texttt{label\_goals}(returns)$
   $(G, D) \leftarrow$
    $\texttt{train\_GAN}(goals, labels, G, D);$
   $goals_{\text{old}} \leftarrow \texttt{update\_replay}(goals)$
**end**

---

Our full algorithm for training a policy $\pi(a_t \mid s_t, g)$ to maximize the coverage objective in Eq. (3) is shown in Algorithm 1. At each iteration $i$, we generate a set of goals by first using $\texttt{sample\_noise}$ to obtain a noise vector $z$ from $p_z(\cdot)$ and then passing this noise to the generator $G$.

We use these goals to train our policy using RL, with the reward function given by Eq. (1) ($\texttt{update\_policy}$). The training can be done with any RL algorithm; in our case we use TRPO (Schulman et al., 2015a) with GAE (Schulman et al., 2015b). Our policy's emperical performance on these goals ($\texttt{evaluate\_policy}$) is used to determine each goal's label $y_g$ ($\texttt{label\_goals}$), as described in Section 4.1. Next, we use these labels to train our goal generator and our goal discriminator ($\texttt{train\_GAN}$), as described in Section 4.2. The generated goals from the previous iteration are used to compute the Monte Carlo estimate of the expectations with respect to the distribution $p_{\text{data}}(g)$ in Eq. (4). By training on goals within $G_i$ produced by the goal generator, our method efficiently finds a policy that optimizes the coverage objective. For details on how we initialize the goal GAN ($\texttt{initialize\_GAN}$), and how we use a replay buffer to prevent "catastrophic forgetting" ($\texttt{update\_replay}$), see Appendix A.

The algorithm described above naturally creates a curriculum for our policy. The goal generator generates goals in $G_i$, for which our policy obtains an intermediate level of return, and thus such goals are at the appropriate level of difficulty for our current policy $\pi_i$. As our policy improves, the generator learns to generate goals in order of increasing difficulty. Hence, our method can be viewed as a way to automatically generate a curriculum of goals. However, the curriculum occurs as a by-product via our optimization, without requiring any prior knowledge of the environment or the tasks that the agent must perform.

## 5 EXPERIMENTAL RESULTS

In this section we provide the experimental results to answer the following questions:

- Does our automatic curriculum yield faster maximization of the coverage objective?
- Does our Goal GAN dynamically shift to sample goals of the appropriate difficulty?
- Does it scale to a higher-dimensional state-space with a low-dimensional space of feasible goals?

To answer the first two questions, we demonstrate our method in two challenging robotic locomotion tasks, where the goals are the $(x, y)$ position of the Center of Mass (CoM) of a dynamically complex quadruped agent. In the first experiment the agent has no constraints and in the second one the agent is inside a U-maze. To answer the third question, we study how our method scales with the

dimension of the state-space in an environment where the feasible region is kept of approximately constant volume in an embedding space that grows in dimension.

We compare our *Goal GAN* method against four baselines. *Uniform Sampling* is a method that does not use a curriculum at all, training at every iteration on goals uniformly sampled from the goal-space. To demonstrate that a straight-forward distance reward can be prone to local minima, *Uniform Sampling with L2 loss* samples goals in the same fashion as the first baseline, but instead of the indicator reward that our method uses, it receives the negative L2 distance to the goal as a reward at every step. We have also adapted two methods from the literature to our setting: Asymmetric Self-play (Sukhbaatar et al., 2017) and SAGG-RIAC (Baranes & Oudeyer, 2013b). Finally, we provide an ablation and an oracle for our method to better understand the importance of sampling "good" goals. The ablation *GAN fit all* consists on not training the GAN only on the "good" goals but rather on every goal attempted in the previous iteration. Given the noise injected at the output of the GAN this generates a gradually expanding set of goals - similar to any hand-designed curriculum. The oracle consists in sampling goals uniformly from the feasible state-space, but only keeping them if they satisfy the criterion defined in Section 4.1. This *Rejection Sampling* method is orders of magnitude more expensive in terms of labeling, but it serves to estimate an upper-bound for our method in terms of performance.

## 5.1 ANT LOCOMOTION

We test our method in two challenging environments of a complex robotic agent navigating either a free space (Free Ant) or a U-shaped maze (Maze Ant). The latter is depicted in Fig. 1, where the orange quadruped is the Ant, and a possible goal to reach is drawn in red. Duan et al. (2016) describe the task of trying to reach the other end of the U-turn and they show that standard RL methods are unable to solve it. We further extend the task to ask to be able to reach any given point within the maze, or the $[-5, 5]^2$ square for Free Ant. The reward is still a sparse indicator function which takes the value 1 only when the $(x, y)$ CoM of the Ant is within $\epsilon = 0.5$ of the goal. Therefore the goal space is 2 dimensional, the state-space is 41 dimensional and the action space is 8 dimensional (see details in Appendix B.1).

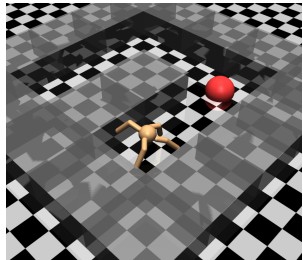

Figure 1: Ant Maze

We first explore whether, by training on goals that are generated by our Goal GAN, we are able to improve our policy's training efficiency, compared to the baselines described above. In Figs. 2a-Fig. 2b we see that our method leads to faster training compared to the baselines. The *Uniform Sampling* baseline does very poorly because too many samples are wasted attempting to train on goals that are infeasible or not reachable by the current policy - hence not receiving any learning signal. If an L2 loss is added to try to guide the learning, the agent falls into a poor local optima of not moving to avoid further negative rewards. The two other baselines that we compare against perform better, but still do not surpass the performance of our method. In particular, Asymmetric Self-play needs to train the goal-generating policy (Alice) at every outer iteration, with an amount of rollouts equivalent to the ones used to train the goal-reaching policy. This additional burden is not represented in the plots, being therefore at least half as sample-efficient as the plots indicate. SAGG-RIAC maintains an ever-growing partition of the goal-space that becomes more and more biased towards areas that already have more sub-regions, leading to reduced exploration and slowing down the expansion of the policy's capabilities. Details of our adaptation of these two methods to our problem, as well as further study of their failure cases, is provided in the Appendices F.1 and F.2.

To better understand the efficiency of our method, we analyze the goals generated by our automatic curriculum. In these Ant navigation experiments, the goal space is two dimensional, allowing us to study the shift in the probability distribution generated by the Goal GAN (Fig. 3) along with the improvement of the policy coverage (Fig. 4). We have indicated the difficulty of reaching the generated goals in Fig. 3. It can be observed in these figures that the location of the generated goals shifts to different parts of the maze, concentrating on the area where the current policy is receiving some learning signal but needs more improvement. The percentage of generated goals that are at the appropriate level of difficulty ("good goals") stays around 20% even as the policy improves. The

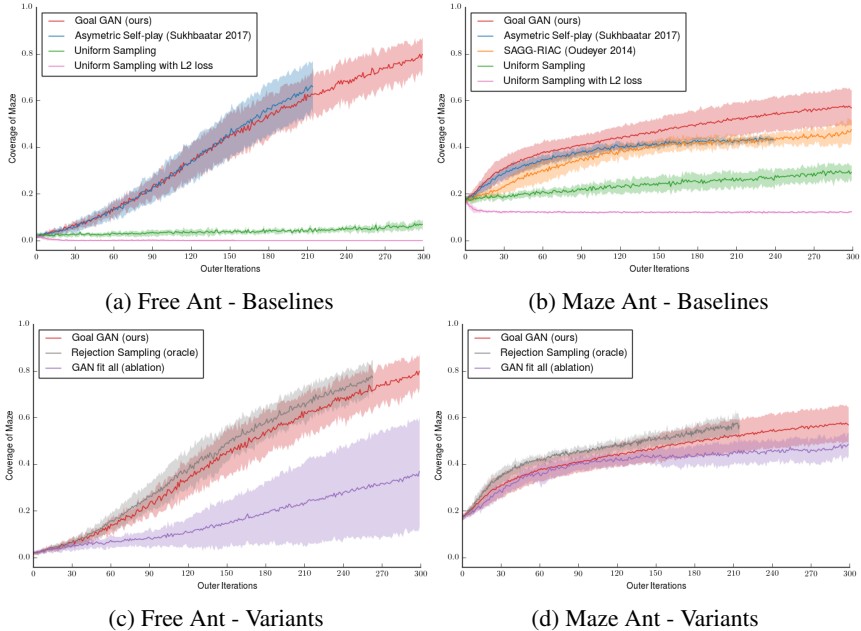

(a) Free Ant - Baselines  (b) Maze Ant - Baselines

(c) Free Ant - Variants  (d) Maze Ant - Variants

Figure 2: Learning curves comparing the training efficiency of our Goal GAN method and different baselines (first row) and variants (second row), for the Free Ant (left column) and the Maze Ant (right column). The y-axis indicates the average return over all feasible goals. The x-axis shows the number of times that new goals have been sampled. All plots average over 10 random seeds.

goals in these figures include a mix of newly generated goals from the Goal GAN as well as goals from previous iterations that we use to prevent our policy from "forgetting" (Appendix A.1). Overall it is clear that our Goal GAN dynamically shift to sample goals of the appropriate difficulty. See Appendix D for additional experiments.

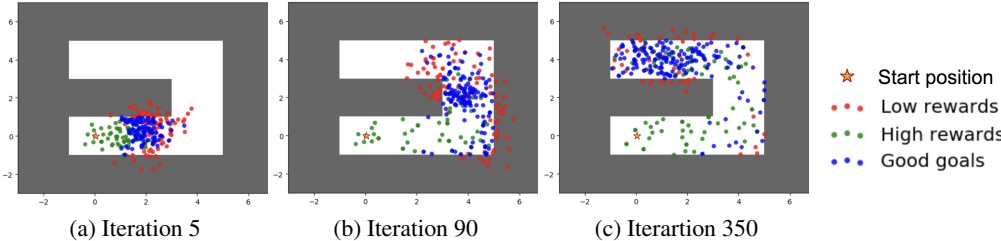

(a) Iteration 5  (b) Iteration 90  (c) Iterartion 350

Figure 3: Goals that our algorithm trains on (200 sampled from the Goal GAN, 100 from the replay). "High rewards" (green) are goals with $\bar{R}^g(\pi_i) \geq R_{\max}$; "Good goals" (blue) have appropriate difficulty for the current policy $R_{\min} \leq \bar{R}^g(\pi_i) \leq R_{\max}$. The red ones have $R_{\min} \geq \bar{R}^g(\pi_i)$

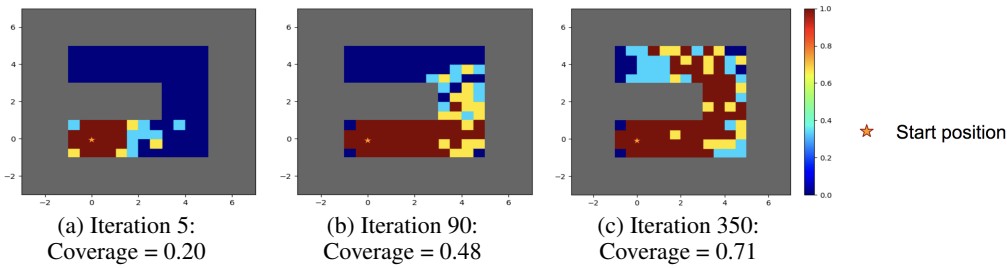

(a) Iteration 5:  (b) Iteration 90:  (c) Iteration 350:
Coverage = 0.20  Coverage = 0.48  Coverage = 0.71

Figure 4: Visualization of the policy performance (same policy training as in Fig. 3). For illustration purposes, the feasible goal-space (i.e. the space within the maze) is divided into a grid, and a goal location is selected from the center of each grid cell. Each grid cell is colored according to the expected return achieved on this goal: Red indicates 100% success; blue indicates 0% success.

It is interesting to analyze the importance of generating "good goals" for efficient learning. This is done in Figs. 2c-2d, where we first show an ablation of our method *GAN fit all*, that disregards the labels. This method performs worse than ours, because the expansion of the goals is not related to the current performance of the policy. Finally, we study the *Rejection Sampling* oracle. As explained in Section 4.1, we wish to sample from the set of "good" goals $G_i$, which we approximate by fitting a Goal GAN to the distribution of good goals observed in the previous policy optimization step. We evaluate now how much this approximation affects learning by comparing the learning performance of our Goal GAN to a policy trained on goals sampled uniformly from $G_i$ by using rejection sampling. This method is orders of magnitude more sample inefficient, but gives us an upper bound on the performance of our method. Figs. 2c-2d demonstrate that our performance is quite close to the performance of this much less efficient baseline.

## 5.2 N-DIMENSIONAL POINT MASS

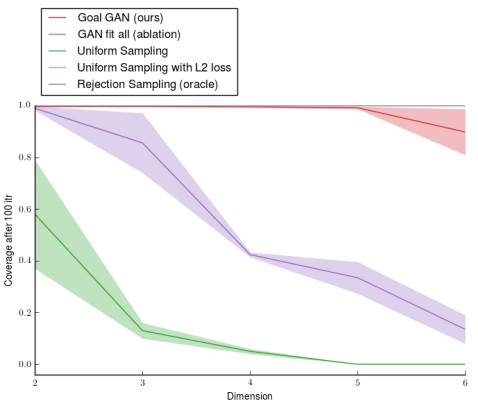

Figure 5: Final goal coverage obtained after 200 outer iterations on the N-dim point mass environment. All plots average over 5 random seeds.

In most real-world RL problems, the set of feasible states is a lower-dimensional subset of the full state space, defined by the constraints of the environment. For example, the kinematic constraints of a robot limit the set of feasible states that the robot can reach. Therefore, uniformly sampling goals from the full state-space would yield very few achievable goals. In this section we use an N-dimensional Point Mass to explore this issue and demonstrate the performance of our method as the embedding dimension increases.

In our experiments, the full state-space of the $N$-dimensional Point Mass is the hypercube $[-5, 5]^N$. However, the Point Mass can only move within a small subset of this state space. In the two-dimensional case, the set of feasible states corresponds to the $[-5, 5] \times [-1, 1]$ rectangle, making up 20% of the full space. For $N > 2$, the feasible space is the Cartesian product of this 2D strip with $[-\epsilon, \epsilon]^{N-2}$, where $\epsilon = 0.3$. In this higher-dimensional environment, our agent receives a reward of 1 when it moves within $\epsilon_N = 0.3\frac{\sqrt{N}}{\sqrt{2}}$ of the goal state, to account for the increase in average $L2$ distance between points in higher dimensions. The ratio of the volume of the embedded space to the volume of the full state space decreases as $N$ increases, down to 0.00023:1 for 6 dimensions.

Fig. 5 shows the performance of our method compared to the other methods, as the number of dimensions increases. The uniform sampling baseline has very poor performance as the number of dimensions increases because the fraction of feasible states within the full state space decreases as the dimension increases. Thus, sampling uniformly results in sampling an increasing percentage of unfeasible states, leading to poor learning signal. In contrast, the performance of our method does not decay as much as the state space dimension increases, because our Goal GAN always generates goals within the feasible portion of the state space (and at the appropriate level of difficulty). The *GAN fit all* variation of our method suffers from the increase in dimension because it is not encouraged to track the narrow feasible region. Finally, the oracle and the baseline with an L2 distance reward have perfect performance, which is expected in this simple task where the optimal policy is just to go in a straight line towards the goal. Even without this prior knowledge, the Goal GAN discovers the feasible subset of the goal space.

## 6 CONCLUSIONS AND FUTURE WORK

We propose a new paradigm in RL where the objective is to train a single policy to succeed on a variety of goals, under sparse rewards. To solve this problem we develop a method for automatic curriculum generation that dynamically adapts to the current performance of the agent. The curriculum is obtained without any prior knowledge of the environment or of the tasks being performed. We use generative adversarial training to automatically generate goals for our policy that are always

at the appropriate level of difficulty (i.e. not too hard and not too easy). In the future we want to combine our goal-proposing strategy with recent multi-goal approaches like HER (Andrychowicz et al., 2017) that could greatly benefit from better ways to select the next goal to train on. Another promising line of research is to build hierarchy on top of the multi-task policy that we obtain with our method by training a higher-level policy that outputs the goal for the lower level multi-task policy (like in Heess et al. (2016) or in Florensa et al. (2017a)). The hierarchy could also be introduced by replacing our current feed-forward neural network policy by an architecture that learns to build implicit plans (Mnih et al., 2016; Tamar et al., 2016), or by leveraging expert demonstrations to extract sub-goals (Zheng et al., 2016), although none of these approaches tackles yet the multi-task learning problem formulated in this work.

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

## A   IMPLEMENTATION DETAILS

### A.1   REPLAY BUFFER

In addition to training our policy on the goals that were generated in the current iteration, we also save a list ("regularized replay buffer") of goals that were generated during previous iterations (`update_replay`). These goals are also used to train our policy, so that our policy does not forget how to achieve goals that it has previously learned. When we generate goals for our policy to train on, we sample two thirds of the goals from the Goal GAN and we sample the one third of the goals uniformly from the replay buffer. To prevent the replay buffer from concentrating in a small portion of goal space, we only insert new goals that are further away than $\epsilon$ from the goals already in the buffer, where we chose the goal-space metric and $\epsilon$ to be the same as the ones introduced in Section 3.1.

### A.2   GOAL GAN INITIALIZATION

In order to begin our training procedure, we need to initialize our goal generator to produce an initial set of goals (`initialize_GAN`). If we initialize the goal generator randomly (or if we initialize it to sample uniformly from the goal space), it is likely that, for most (or all) of the sampled goals, our initial policy would receives no reward due to the sparsity of the reward function. Thus we might have that all of our initial goals $g$ have $\bar{R}^g(\pi_0) < R_{\min}$, leading to very slow training.

To avoid this problem, we initialize our goal generator to output a set of goals that our initial policy is likely to be able to achieve with $\bar{R}^g(\pi_i) \geq R_{\min}$. To accomplish this, we run our initial policy $\pi_0(a_t \mid s_t, g)$ with goals sampled uniformly from the goal space. We then observe the set of states $S^v$ that are visited by our initial policy. These are states that can be easily achieved with the initial policy, $\pi_0$, so the goals corresponding to such states will likely be contained within $S_0^I$. We then train the goal generator to produce goals that match the state-visitation distribution $p_v(g)$, defined as the uniform distribution over the set $f(S^v)$. We can achieve this through traditional GAN training, with $p_{\text{data}}(g) = p_v(g)$. This initialization of the generator allows us to bootstrap the Goal GAN training process, and our policy is able to quickly improve its performance.

## B   EXPERIMENTAL DETAILS

### B.1   ANT SPECIFICATIONS

The ant is a quadruped with 8 actuated joints, 2 for each leg. The environment is implemented in Mujoco (Todorov et al., 2012). Besides the coordinates of the center of mass, the joint angles and joint velocities are also included in the observation of the agent. The high degrees of freedom make navigation a quite complex task requiring motor coordination. More details can be found in Duan et al. (2016), and the only difference is that in our goal-oriented version of the Ant we append the observation with the goal, the vector from the CoM to the goal and the distance to the goal. For the Free Ant experiments the objective is to reach any point in the square $[-5m, 5m]^2$ on command. The maximum time-steps given to reach the current goal are 500.

### B.2   ANT MAZE ENVIRONMENT

The agent is constrained to move within the maze environment, which has dimensions of 6m x 6m. The full state-space has an area of size 10 m x 10 m, within which the maze is centered. To compute the coverage objective, goals are sampled from within the maze according to a uniform grid on the maze interior. The maximum time-steps given to reach the current goal are 500.

### B.3   POINT-MASS SPECIFICATIONS

For the N-dim point mass of Section 5.2, in each episode (rollout) the point-mass has 400 timesteps to reach the goal, where each timestep is 0.02 seconds. The agent can accelerate in up to a rate of 5 $m/s^2$ in each dimension ($N = 2$ for the maze). The observations of the agent are $2N$ dimensional, including position and velocity of the point-mass.

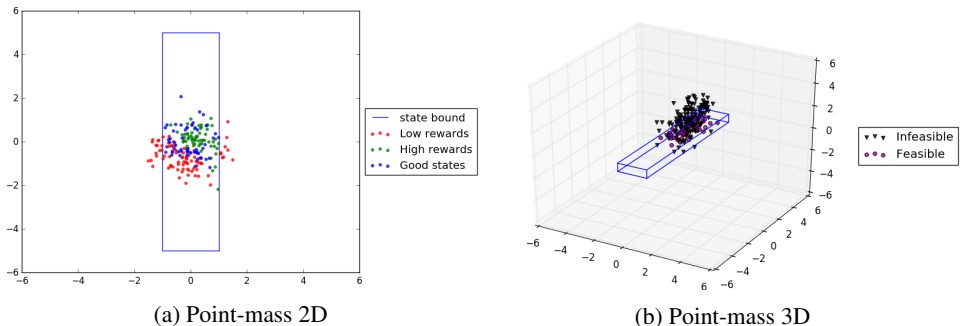

(a) Point-mass 2D  (b) Point-mass 3D

Figure 6: Representation of the point-mass experiments with $N = 2$ and $N = 3$. The points contained within the blue boundaries are the feasible goals, corresponding to the positions that can be reached.

### B.4 GOAL GAN DESIGN AND TRAINING

After the generator generates goals, we add noise to each dimension of the goal sampled from a normal distribution with zero mean and unit variance. At each step of the algorithm, we train the policy for 5 iterations, each of which consists of 100 episodes. After 5 policy iterations, we then train the GAN for 200 iterations, each of which consists of 1 iteration of training the discriminator and 1 iteration of training the generator. The generator receives as input 4 dimensional noise sampled from the standard normal distribution. The goal generator consists of two hidden layers with 128 nodes, and the goal discriminator consists of two hidden layers with 256 nodes, with relu nonlinearities.

### B.5 POLICY AND OPTIMIZATION

The policy is defined by a neural network which receives as input the goal appended to the agent observations described above. The inputs are sent to two hidden layers of size 32 with tanh non-linearities. The final hidden layer is followed by a linear $N$-dimensional output, corresponding to accelerations in the $N$ dimensions. For policy optimization, we use a discount factor of 0.998 and a GAE lambda of 0.995. The policy is trained with TRPO with Generalized Advantage Estimation implemented in rllab (Schulman et al., 2015a;b; Duan et al., 2016). Every "update_policy" consists of 5 iterations of this algorithm.

## C STUDY OF GOALGAN "GOOD" GOALS

To label a given goal (Section 4.1), we could empirically estimate the expected return for this goal $\bar{R}^g(\pi_i)$ by performing rollouts of our current policy $\pi_i$. The label for this goal is then set to $y_g = \mathbb{1}\left\{R_{\min} \leq \bar{R}^g(\pi_i) \leq R_{\max}\right\}$. Nevertheless, having to execute additional rollouts just for labeling is not sample efficient. Therefore, we instead use the rollouts that were used for the most recent policy update. This is an approximation as the rollouts where performed under $\pi_{i-1}$, but as we show in Figs. 7a-7b, this small "delay" does not affect learning significantly. Indeed, using the true label (estimated with three new rollouts from $\pi_i$) yields the *Goal GAN true label* curves that are only slightly better than what our method does.

In the same plots we also study another definition of "good" goals that has been previously used in the literature: learning progress (Baranes & Oudeyer, 2013b; Graves et al., 2017). Given that we work in a continuous goal-space, estimating the learning progress of a single goal requires estimating the performance of the policy on that goal before the policy update and after the policy update (potentially being able to replace one of these estimations with the rollouts from the policy optimization, but not both). Therefore the method does require more samples, but we deemed interesting to compare how well the metric to automatically build a curriculum. We see in the Figs. 7a-7b that the two metrics yield a very similar learning, at least in the case of Ant navigation tasks with sparse rewards.

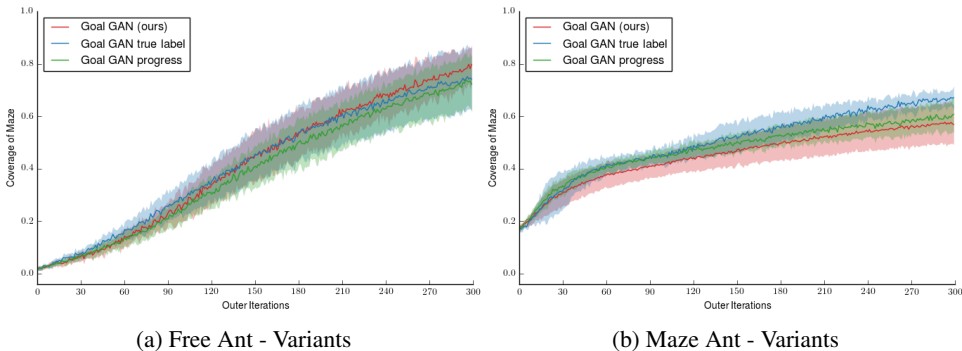

(a) Free Ant - Variants                    (b) Maze Ant - Variants

Figure 7: Learning curves comparing the training efficiency of our method and different variants. All plots are an average over 10 random seeds.

## D   GOAL GENERATION FOR FREE ANT

Similar to the experiments in Figures 3 and 4, here we show the goals that were generated for the Free Ant experiment in which a robotic quadruped must learn to move to all points in free space. Figures 8 and 9 show the results. As shown, our method produces a growing circle around the origin; as the policy learns to move the ant to nearby points, the generator learns to generate goals at increasingly distant positions.

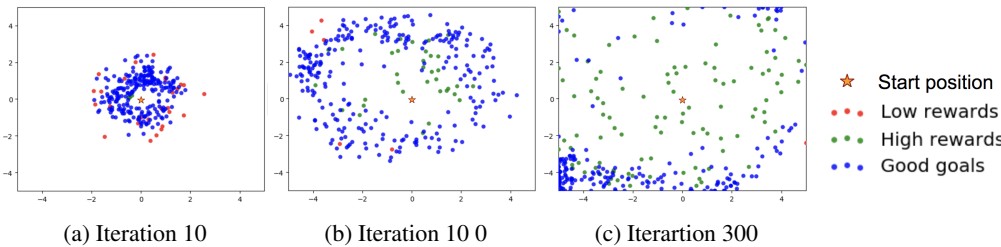

(a) Iteration 10          (b) Iteration 10 0        (c) Iterartion 300

Figure 8: Goals that our algorithm trains on (200 sampled from the Goal GAN, 100 from the replay). "High rewards" (green) are goals with $\bar{R}^g(\pi_i) \geq R_{\max}$; "Good goals" (blue) have appropriate difficulty for the current policy $R_{\min} \leq \bar{R}^g(\pi_i) \leq R_{\max}$. The red ones have $R_{\min} \geq \bar{R}^g(\pi_i)$

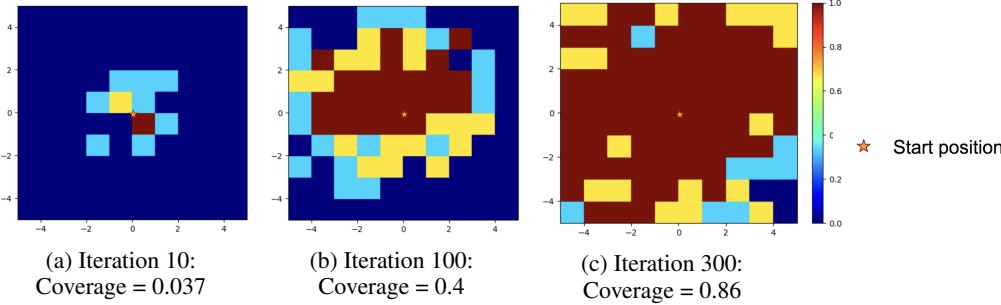

(a) Iteration 10:         (b) Iteration 100:       (c) Iteration 300:
Coverage = 0.037       Coverage = 0.4        Coverage = 0.86

Figure 9: Visualization of the policy performance for different parts of the state space (same policy training as in Fig. 8). For illustration purposes, the feasible state-space is divided into a grid, and a goal location is selected from the center of each grid cell. Each grid cell is colored according to the expected return achieved on this goal: Red indicates 100% success; blue indicates 0% success.

## E   MULTI-PATH POINT-MASS MAZE

In this section we show that our Goal GAN method is efficient at tracking clearly multi-modal distributions of good goals. To this end, we introduce a new maze environment with multiple paths,

as can be seen in Fig. 10. To keep the experiment simple we replace the Ant agent by a point-mass environment (in orange), which actions are directly the velocity vector (2 dim). As in the other experiments, our aim is to learn a policy that can reach any feasible goal corresponding to $\epsilon$-balls in state space like the one depicted in red.

Similar to the experiments in Figures 3 and 4, here we show the goals that were generated for the Mutli-path point-mass maze experiment. Figures 11 and 12 show the results. It can be observed that our method produces a multi-modal distribution over goals, tracking all the areas where goals are at the appropriate level of difficulty. Note that the samples from the regularized replay buffer are responsible for the trailing spread of "High Reward" goals and the Goal GAN is responsible for the more concentrated nodes, as can be seen in Fig. 13. A clear benefit of using our Goal GAN as a generative model is that no prior knowledge about the distribution to fit is required (like the number of modes). Finally, note that the fact of having several possible paths to reach a specific goal does not hinder the learning of our algorithm that consistently reaches full coverage in this problem as seen in Fig. 14.

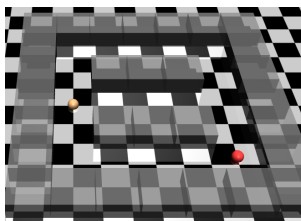

Figure 10: Multi-path Point-mass Maze

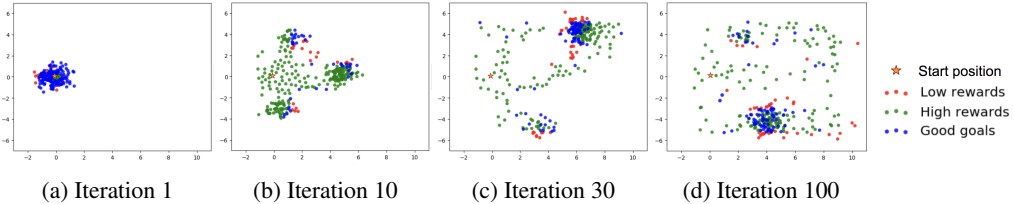

(a) Iteration 1      (b) Iteration 10      (c) Iteration 30      (d) Iteration 100

Figure 11: Goals that our algorithm trains on (200 sampled from the Goal GAN, 100 from the replay). "High rewards" (green) are goals with $\bar{R}^g(\pi_i) \geq R_{\max}$; "Good goals" (blue) have appropriate difficulty for the current policy $R_{\min} \leq \bar{R}^g(\pi_i) \leq R_{\max}$. The red ones have $R_{\min} \geq \bar{R}^g(\pi_i)$

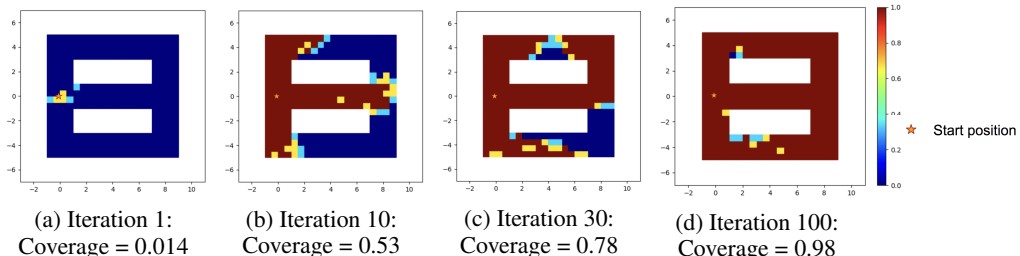

(a) Iteration 1: Coverage = 0.014    (b) Iteration 10: Coverage = 0.53    (c) Iteration 30: Coverage = 0.78    (d) Iteration 100: Coverage = 0.98

Figure 12: Visualization of the policy performance for different parts of the state space (same policy training as in Fig. 8). For illustration purposes, the feasible state-space is divided into a grid, and a goal location is selected from the center of each grid cell. Each grid cell is colored according to the expected return achieved on this goal: Red indicates 100% success; blue indicates 0% success.

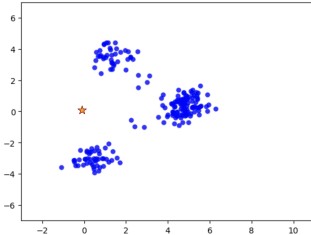

Figure 13: Iteration 10 Goal GAN samples (Fig. 11b without replay samples)

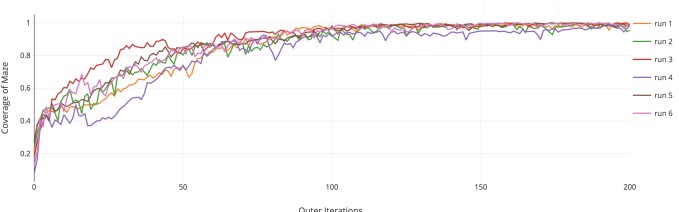

Figure 14: Learning curves of our algorithm on Multi-path Point-mass Maze, consistently achieving full coverage

# F COMPARISONS WITH OTHER METHODS

## F.1 ASYMMETRIC SELF-PLAY (SUKHBAATAR ET AL., 2017)

Although not specifically designed for the problem presented in this paper, it is straight forward to apply the method proposed by Sukhbaatar et al. (2017) to our problem. An interesting study of its limitations in a similar setting can be found in (Florensa et al., 2017b).

## F.2 SAGG-RIAC (BARANES & OUDEYER, 2013B)

In our implementation of this method, we use TRPO as the "Low-Level Goal-Directed Exploration with Evolving Context". We therefore implement the method as batch: at every iteration, we sample $N_{new}$ new goals $\{y_i\}_{i=0...N_{new}}$, then we collect rollouts of $t_{max}$ steps trying to reach them, and perform the optimization of the parameters using all the collected data. The detailed algorithm is given in the following pseudo-code.

---

**Algorithm 2:** Generative Goal with Sagg-RIAC

**Hyperparameters:** window size $\zeta$, tolerance threshold $\epsilon_{max}$, competence threshold $\epsilon_C$, maximum time horizon $t_{max}$, number of new goals $N_{new}$, maximum number of goals $g_{max}$, mode proportions $(p_1, p_2, p_3)$;
**Input** : Policy $\pi_{\theta_0}(s_{start}, y_g)$, goal bounds $B_Y$, reset position $s_{rest}$
**Output:** Policy $\pi_{\theta_N}(s_{start}, y_g)$
$\mathbf{R} \leftarrow \{(R_0, \Gamma_{R_0})\}$ where $R_0 = Region(B_Y), \Gamma_{R_0} = 0$;
**for** $i \leftarrow 1$ **to** $N$ **do**
    $goals \leftarrow$ **Self-generate** $N_{new}$ goals: $\{y_j\}_{j=0...N_{new}}$;
    $paths = [\,]$;
    **while** $number\_steps\_in(paths) < batch\_size$ **do**
        Reset $s_0 \leftarrow s_{rest}$;
        $y_g \leftarrow \text{Uniform}(goals)$;
        $y_f, \Gamma_{y_g}, path \leftarrow \texttt{collect\_rollout}(\pi_{\theta_i}(\cdot, y_g), s_{reset})$;
        $paths.\texttt{append}(path)$;
        **UpdateRegions(R,** $y_f, 0)$ ;
        **UpdateRegions(R,** $y_g, \Gamma_{y_g})$;
    **end**
    $\pi_{\theta_{i+1}} \leftarrow$ train $\pi_{\theta_i}$ with TRPO on collected $paths$;
**end**

---

**UpdateRegions(R,** $y_f, \Gamma_{y_f})$ is exactly the Algorithm 2 described in the original paper, and **Self-generate** is the "Active Goal Self-Generation (high-level)" also described in the paper (Section 2.4.4 and Algorithm 1), but it's repeated $N_{new}$ times to produce a batch of $N_{new}$ goals jointly. As for the competence $\Gamma_{y_g}$, we use the same formula as in their section 2.4.1 (use highest competence if reached close enough to the goal) and $C(y_g, y_f)$ is computed with their equation (7). The `collect_rollout` function resets the state $s_0 = s_{reset}$ and then applies actions following the goal-conditioned policy $\pi_\theta(\cdot, y_g)$ until it reaches the goal or the maximum number of steps $t_{max}$ has been taken. The final state, transformed in goal space, $y_f$ is returned.

As hyperparameters, we have used the recommended ones in the paper, when available: $p_1 = 0.7$, $p_2 = 0.2$, $p_3 = 0.1$. For the rest, the best performance in an hyperparameter sweep yields: $\zeta = 100$, $g_{max} = 100$. The noise for mode(3) is chosen to be Gaussian with variance 0.1, the same as the tolerance threshold $\epsilon_{max}$ and the competence threshold $\epsilon_C$.

As other details, in our tasks there are no constraints to penalize for, so $\rho = \emptyset$. Also, there are no sub-goals. The reset value $r$ is 1 as we reset to $s_{start}$ after every reaching attempt. The number of explorative movements $q \in \mathbb{N}$ has a less clear equivalence as we use a policy gradient update with a stochastic policy $\pi_\theta$ instead of a SSA-type algorithm.

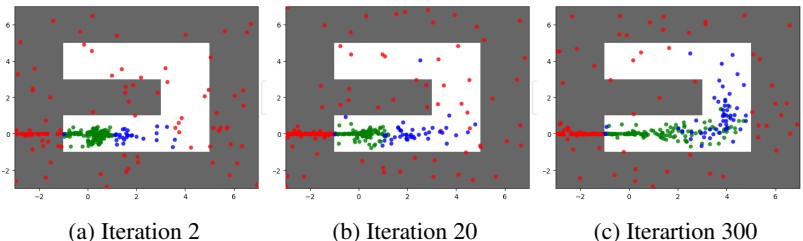

(a) Iteration 2     (b) Iteration 20     (c) Iterartion 300

Figure 15: Goals sampled by SAGG-RIAC (same policy training as in Fig. 16). "High rewards" (in green) are goals with $\bar{R}^g(\pi_i) \geq R_{\max}$; "Good goals" (in blue) are those with the appropriate level of difficulty for the current policy ($R_{\min} \leq \bar{R}^g(\pi_i) \leq R_{\max}$). The red ones have $R_{\min} \geq \bar{R}^g(\pi_i)$

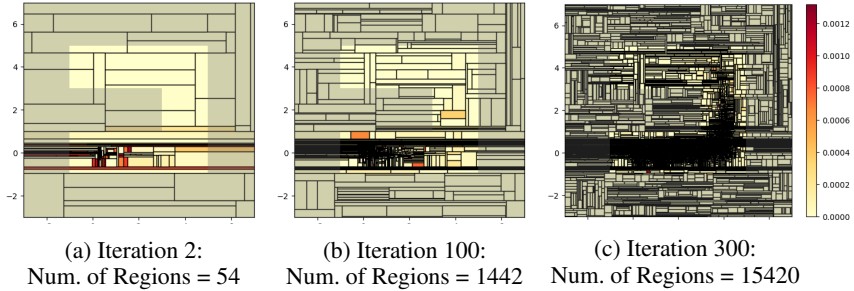

(a) Iteration 2:     (b) Iteration 100:     (c) Iteration 300:
Num. of Regions = 54 Num. of Regions = 1442 Num. of Regions = 15420

Figure 16: Visualization of the regions generated by the SAGG-RIAC algorithm

