# OpenReview forum: "Automatic Goal Generation for Reinforcement Learning Agents"
_ICLR.cc/2018/Conference — Reject_

### Official Review · AnonReviewer2 · 2017-11-26
**well-written paper, useful addition to literature, doubts about stability**

**Rating:** 8
**Confidence:** 4

**Review:**

In general I find this to be a good paper and vote for acceptance. The paper is well-written and easy to follow.  The proposed approach is a useful addition to existing literature.

Besides that I have not much to say except one point I would like to discuss:

In 4.2 I am not fully convinced of using an adversial model for goal generation. RL algorithms generally suffer from poor stability  and GANs themselves can have convergence issues. This imposes another layer of possible instability.

Besides, generating useful reward function, while not trivial, can be seen as easier than solving the full RL problem.
Can the authors argue why this model class was chosen over other, more simple, generative models?
Furthermore, did the authors do experiments with simpler models?

Related:
"We found that the LSGAN works better than other forms of GAN for our problem."
Was this improvement minor, or major, or didn't even work with other GAN types? This question is important, because for me the big question is if this model is universal and stable in a lot of applications or requires careful fine-tuning and monitoring.

---
Update:
The authors addressed  the major point of criticism in my review.  I am now more convinced in the quality of the proposed work, and have updated my review score accordingly.

---

> ### Author Response · Authors · 2017-12-27
> **Thank you for recognizing the contribution in this paper and comment on GAN training stability.**
>
> Thank you for recognizing the contribution in this paper.  We agree that care must be taken to ensure stability for training the GAN.  Still, our experiments show that our method outperforms the competing approaches on this problem.  We chose to use a GAN rather than another generative model due to a GAN’s demonstrated ability to generate samples in high-dimensional spaces (such as images), thus giving our method the potential to scale up to high-dimensional goal spaces.  We did not experiment with other generative models for these tasks.
>
> Regarding a comparison of different GAN types: in our experiments, using a WGAN (Arjovsky et al. 2017) led to significantly more stable training than a vanilla GAN (as in Goodfellow et al., 2014), and using an LSGAN improved the training stability even further, but not quite as dramatically. We have added these observations in the paper without additional details as it is not the focus of our work. As is stated in Section 4.2, all results shown in our paper, across a number of different environments, use the LSGAN with the original hyperparameters reported in Mao et al. 2017. In general, we’ve found GANs to be much more stable in lower dimensional state spaces than in image spaces, and many of the well known convergence issues did not happen. Therefore, no considerable fine-tuning and monitoring was needed. In future work we hope to extend our model to an even greater number of environments.

---

> > ### Comment · AnonReviewer2 · 2018-01-23
> > **Clarification**
> >
> > Thanks for the reply! Could you maybe clarify:
> >
> > "WGAN (Arjovsky et al. 2017) led to significantly more stable training than a vanilla GAN (as in Goodfellow et al., 2014), and using an LSGAN improved the training stability even further, but not quite as dramatically"
> >
> > I am not so much interested in which GAN works the best, also not interested in minor performance improvements. What I am interested in is how justified my doubts about possible additional instability  in the algorithm are. So, let me rephrase a bit the question:
> >
> > 1.  When you chose to use a WGAN, was it because a standard GAN did not work reliably? Or are we talking about merely incremental improvements?
> >
> > In essence, which  of the following two statements is more true:
> >
> > a) When using GAN for automatic goal generation it is essential to tune the hyperparameter (such as which loss function is used (standard, WGAN, etc), for it to be robust.
> >
> > b) Our approach does work reliably with any type of adversarial method (and you made some good arguments why to use them in this situation).   To show the best results,  we used WGAN in our experiments.

---

> > > ### Author Response · Authors · 2018-01-24
> > > **Our method should work with any GAN powerful enough to capture the desired goal distribution.**
> > >
> > > The statement b) is more true. Our method should work with any GAN powerful enough to capture the desired goal distributions. However, we did observe that some of the GAN methods are more stable than others, possibly due to the fact that they have less hyper-parameters to tune. Our GAN hyper-parameter tuning was *not* done in a per environment basis and our tuned hyper-parameters were shared across all the experiments. Therefore, due to computation limit, we chose to report the results with LSGAN that has the least number of hyper-parameters to tune (and in fact the hyper-parameters from the original paper worked fine, and we did not try others).
> > > More precisely, when trying the other GAN methods, they would sometimes fit less accurately one "good goals" distribution (probably improvable with better initial tuning). This generates more goals with too high or too low rewards, therefore momentarily decreasing the learning efficiency of our algorithm, and taking longer to solve the task. In other words, our approach works reliably with any GAN able to generate some new samples from the distribution it is fitting (all the ones we tried satisfy this - as any proper generative model should), and the performance of the algorithm increases with how well it fits the distributions.

---

### Official Review · AnonReviewer3 · 2017-11-27
**Interesting problem, but approach / results are not completely clear.**

**Rating:** 4
**Confidence:** 4

**Review:**

Summary:

This paper proposes to use a GAN to generate goals to implement a form of curriculum learning. A goal is defined as a subset of the state space. The authors claim that this model can discover all "goals" in the environment and their 'difficulty', which can be measured by the success rate / reward of the policy. Hence the goal network could learn a form of curriculum, where a goal is 'good' if it is a state that the policy can reach after a (small) improvement of the current policy.

Training the goal GAN is done via labels, which are states together with the achieved reward by the policy that is being learned.

The benchmark problems are whether the GAN generates goals that allow the agent to reach the end of a U-maze, and a point-mass task.

Authors compare GAN goal generation vs uniformly choosing a goal and 2 other methods.

My overall impression is that this work addresses an interesting question, but the experimental setup / results are not clearly worked out. More broadly, the paper does not address how one can combine RL and training a goal GAN in a stable way.

Pro:
- Developing hierarchical learning methods to improve the sample complexity of RL is an important problem.
- The paper shows that the U-maze can be 'solved' using a variety of methods that generate goals in a non-uniform way.

Con:
- It is not clear to me how the asymmetric self-play and SAGG-RIAC are implemented and why they are natural baselines.
- It is not clear to me what the 'goals' are in the point mass experiment. This entire experiment should be explained much more clearly (+image).
- It is not clear how this method compares qualitatively vs baselines (differences in goals etc).
- This method doesn't seem to always outperform the asymm-selfplay baseline. The text mentions that baseline is less efficient, but this doesn't make the graph very interpretable.
- The curriculum in the maze-case consists of regions that just progress along the maze, and hence is a 1-dimensional space. Hence using a manually defined set of goals should work quite well. It would be better to include such a baseline as well.
- The experimental maze-setting and point-mass have a simple state / goal structure. How can this method generalize to harder problems?
-- The entire method is quite complicated (e.g. training GANs can be highly unstable). How do we stabilize / balance training the GAN vs the RL problem?
-- I don't see how this method could generalize to problems where the goals / subregions of space do not have a simple distribution as in the maze problem, e.g. if there are multiple ways of navigating a maze towards some final goal state. In that case, to discover a good solution, the generated goals should focus on one alternative and hence the GAN should have a unimodal distribution. How do you force the GAN in a principled way to focus on one goal in this case? How could you combine RL and training the GAN stably in that case?

Detailed:
- (2) is a bit strange: shouldn't the indicator say: 1( \exists t: s_t \in S^g )? Surely not all states in the rollout (s_0 ... s_t) are in the goal subspace: the indicator does not factorize over the union. Same for other formulas that use \union.
- Are goals overlapping or non-overlapping subsets of the state space?
Definition around (1) basically says it's non-overlapping, yet the goal GAN seems to predict goals in a 2d space, hence the predicted goals are overlapping?
- What are the goals that the non-uniform baselines predict? Does the GAN produce better goals?
- Generating goal labels is
- Paper should discuss literature on hierarchical methods that use goals learned from data and via variational methods:
1. Strategic Attentive Writer (STRAW), V. Mnih et al, NIPS 2016
2. Generating Long-term Trajectories Using Deep Hierarchical Networks. S.
Zheng et al, NIPS 2016

---

> ### Author Response · Authors · 2017-12-27
> **Answer to: “It is not clear to me how the asymmetric self-play and SAGG-RIAC are implemented and why they are natural baselines.”**
>
> Our implementation of “Asymmetric Self-Play” follows directly from the description of their method from their publication.  In Asymmetric Self-play, “Alice” proposes goals (exactly what our Goal GAN does) for the agent “Bob” to try to achieve, and Alice and Bob are both trained with reinforcement learning (we use TRPO, with the same parameters as for our method).  We use the “repeat” version of asymmetric self-play in which “Bob” must then learn to reach the goal that “Alice” proposed.  In the Asymmetric Self-play paper, training is alternated between a “multi-goal” setup and a single “target task” setup.  In our case we do not alternate because our “target task” setup is the same as the “multi-goal” one: we desire to train an agent that can achieve many target tasks, which is already done by the multi-goal setup; thus we only need the “multi-goal” training portion of their method.  Their multi-goal training method, if successful, would result in a policy in which “Bob” learns to achieve many goals.  Since this is also the objective of our method (described in equation 3 of our paper), Asymmetric Self-play is an appropriate baseline for our task.
>
> Regarding SAGG-RIAC, details of our implementation of this method can be found in Appendix E.2.  The objective of SAGG-RIAC is the same as the objective of our method, although SAGG-RIAC is usually used to train a model-based agent whereas our method also works with an agent trained in a model-free setting.  Regardless, since SAGG-RIAC likewise attempts to train an agent to achieve many goals, it is also a natural baseline to compare against.

---

> ### Author Response · Authors · 2017-12-27
> **Answer to: "It is not clear to me what the 'goals' are in the point mass experiment. This entire experiment should be explained much more clearly (+image).”**
>
> The goals are simply points in n-dimensional space.  The purpose of this experiment is to evaluate how well our method scales up to goals of higher dimensions.  Thus the environment places an n-dimensional point-mass in an n-dimensional space in which the point mass is constrained to move within a small region within this space.  The feasible goals are points within this smaller region, and the agent achieves a goal by moving to within epsilon of the goal.  The difficulty of this problem for goal-generation is that the goal-generator must learn to discover the bounds of the smaller region within which the agent is constrained to move.  Finding this region becomes increasingly challenging as the dimensionality of the state space increases.  Our goal generation method bootstraps from states visited by the agent and thus is able to efficiently find this feasible region.

---

> ### Author Response · Authors · 2017-12-27
> **Answer to: “It is not clear how this method compares qualitatively vs baselines (differences in goals etc).”**
>
> The Asymmetric Self-play method is also used as baseline in other task-generation papers (Florensa et al., 2017), where we can find a comprehensive analysis of the generation process of Asymmetric Self-play. We summarize here the most relevant findings in this other work. Asymmetric Self-Play relies on an agent “Alice” proposing goals.  However, in a continuous action space, Alice is typically represented as a unimodal Gaussian policy.  Thus, rather than proposing a diverse set of goals, Alice will tend to propose goals in a small cluster around the mean of the Gaussian that represents Alice’s policy.  In contrast, our Goal GAN can produce goals to match an arbitrary goal distribution, giving our method much more flexibility and leading to improved performance.
>
> Furthermore, because Asymmetric Self-play uses a goal generation agent (“Alice”) that is trained with reinforcement learning, the goal generator can suffer from the problem of sparse rewards when Bob makes a large improvement relative to Alice.  This instability is also described in (Florensa et al., 2017).
>
> The goals generated by SAGG-RIAC can be seen in Figures 9 and 10 in the appendix of our paper.  As explained in Section 5.1 of our paper, “SAGG-RIAC maintains an ever-growing partition of the goal-space that becomes more and more biased towards areas that already have more sub-regions, leading to reduced exploration and slowing down the expansion of the policy’s capabilities.”

---

> ### Author Response · Authors · 2017-12-27
> **Answer to: “This method doesn't seem to always outperform the asymm-selfplay baseline. The text mentions that baseline is less efficient, but this doesn't make the graph very interpretable.”**
>
> Our method consistently outperforms the Asymmetric Self-Play baseline.  This is not currently properly reflected in our graphs, since the Asymmetric Self-Play baseline requires extra rollouts to train “Alice” that are not currently included in our plots.  In the final version of our paper, we will include the Alice rollouts in our plot to make this more clear.  Due to the extra rollouts needed to train Alice, our method is much more sample efficient than this baseline.

---

> ### Author Response · Authors · 2017-12-27
> **Answer to: “The curriculum in the maze-case consists of regions that just progress along the maze, and hence is a 1-dimensional space. Hence using a manually defined set of goals should work quite well. It would be better to include such a baseline as well.”**
>
> This baseline would, unfortunately, only work for this one task, whereas our method is more general and also works for the other tasks shown in our paper (e.g. Free Ant, N-dimensional Point Mass).  Another difficulty with this approach would be to choose at what rate to increment the generated goals along the maze (i.e. at what rate to progress the curriculum).  In contrast, our method uses the performance of the policy to automatically determine which goals are generated at each time step.

---

> ### Author Response · Authors · 2017-12-27
> **Answer to: “The experimental maze-setting and point-mass have a simple state / goal structure. How can this method generalize to harder problems?”**
>
> In this paper, we evaluate our method compared to existing baselines for the topic of multi-task goal generation and found that our method outperforms previous competing approaches.  Our paper thus establishes our method as a promising direction for multi-task goal generation which can be extended to other tasks in future work. Furthermore, GANs have been shown to be a powerful framework to generate samples from considerably higher dimensional and complex distributions, such as images. Therefore, we think our method has more potential than others to properly generalize to harder goal structures.

---

> ### Author Response · Authors · 2017-12-27
> **Answer to: “The entire method is quite complicated (e.g. training GANs can be highly unstable). How do we stabilize / balance training the GAN vs the RL problem?”**
>
> The training of the Goal GAN and the training of the RL agent is balanced / stabilized through their connected objective, in which the Goal GAN is trained to generate goals for which the RL agent obtains an intermediate level of return (Section 4.1).  The Goal GAN is trained using labels indicating, for each goal, whether the RL agent can obtain an intermediate level of return for that goal.  These labels are computed empirically from rollouts collected by the RL agent.  Thus, if the RL agent’s performance is slowly increasing, then the goals that the Goal GAN produces will remain relatively similar across timesteps, whereas if the performance of the RL agent increases dramatically, then the Goal GAN will quickly adjust the goals that it is generating to generate goals that are at the appropriate level of difficulty for the current policy.  The shared objective ensures that the Goal GAN always generates goals that are appropriate for the RL agent at each iteration.

---

> ### Author Response · Authors · 2017-12-27
> **Answer to: “I don't see how this method could generalize to problems where the goals [...] How do you force the GAN in a principled way to focus on one goal in this case? How could you combine RL and training the GAN stably in that case?”**
>
> The purpose of the Goal GAN is to generate all feasible goals within a state space (at the appropriate rate based on the performance of the RL agent).  If there are multiple paths through a maze, then the Goal GAN should eventually generate goals at all states along all such paths.  For example, see Figure 7 in the appendix, in which an ant in free space learns to move in many possible directions; the generated goals form a circle that grows outward from the initial position of the ant.  In such a case, the RL agent is trained to reach each of these different goal locations.  Thus, the case in which there are multiple paths to achieve each goal does not present any problems for our method.

---

> ### Author Response · Authors · 2017-12-27
> **Answer to: “(2) is a bit strange: shouldn't the indicator say: 1( \exists t: s_t \in S^g )? Surely not all states in the rollout (s_0 ... s_t) are in the goal subspace: the indicator does not factorize over the union. Same for other formulas that use \union.”**
>
> The “union”-like operator in this expression is intended to indicate the OR operation, e.g. the expression in 2 expands to:
> Indicator(s_0 is in S_g OR s_1 is in S_g OR … OR s_T is in S_g)
> We will make this clear in the final version of our paper.

---

> ### Author Response · Authors · 2017-12-27
> **Answer to: “Are goals overlapping or non-overlapping subsets of the state space? Definition around (1) basically says it's non-overlapping, yet the goal GAN seems to predict goals in a 2d space, hence the predicted goals are overlapping?”**
>
> Goals are overlapping subsets of the state space.  Thus a single state may be contained in multiple goal sets $S^g$.  Our RL agent receives only a single goal as input at a time, so this case does not cause any problems for our method.

---

> ### Author Response · Authors · 2017-12-27
> **Answer to: “What are the goals that the non-uniform baselines predict? Does the GAN produce better goals?”**
>
> See the below discussion on this topic. One can qualitatively compare between the Goal GAN generated goals in Fig. 2 and the SAGG-RIAC ones in Fig. 9.

---

> ### Author Response · Authors · 2017-12-27
> **Answer to: “Generating goal labels is”**
>
> This sentence seems to be incomplete.  The reviewer is invited to re-submit this comment if it has not been answered by our response.

---

> ### Author Response · Authors · 2017-12-27
> **Answer to: “Paper should discuss literature on hierarchical methods that use goals learned from data and via variational methods: 1. STRAW, V. Mnih et al, NIPS 2016 2. Generating Long-term Trajectories Using Deep Hierarchical Networks. S. Zheng et al, NIPS 2016”**
>
> We thank the reviewer for these references and we will discuss them in our related work section. None of the referenced literature directly tackles the multi-task problem solved by our proposed method, but they are complementary. Neither of them allows to condition the overall policy on different goals (the “action-plans“ in STRAW or the “macro-goals” in HPN are internals of the policy, not an input that can be changed externally). In fact, HPN is only used in a supervised setting trying to imitate expert trajectories - which is weakly related to our problem where no demonstrations are required. Our trained policy does not have any explicit hierarchy like the ones proposed in these papers, which makes it orthogonal to them - and also complementary! It would be an interesting research to improve our approach by learning a hierarchical policy instead the MLP used in our experiments (as described in Appendix B.5). This goes beyond the scope of the current paper and is left as future work.

---

### Official Review · AnonReviewer1 · 2017-11-27
**Review for Automatic Goal Generation for Reinforcement Learning Agents**

**Rating:** 6
**Confidence:** 4

**Review:**

This paper proposed a method for automatic curriculum generation that allow an agent to learn to reach multiple goals in an environment with considerable sample efficiency. They use a generator network to propose tasks for the agent accomplish. The generator network is trained with GAN.  In addition, the proposed method is also shown to be able to solve tasks with sparse rewards without the need manually modify reward functions. They compare the Goal GAN method with four baselines, including Uniform sampling, Asymmetric Self-play, SAGG-RIAC, and Rejection sampling. The proposed method is tested on two environments: Free Ant and Maze Ant. The empirical study shows that the proposed method is able to improve policies’ training efficiency comparing to these baselines. The technical contributions seem sound, however I find it is slightly difficult to fully digest the whole paper without getting the insight from each individual piece and there are some important details missing, as I will elaborate more below.

1. it is unclear to me why the proposed method is able to solve tasks with sparse rewards? Is it because of the horizons of the problems considered are not long enough? The author should provide more insight for this contribution.

2. It is unclear to me how R_min and R_max as hyperparameters are obtained and how their settings affect the performance.

3. Another concern I have is regarding the generalizability of the proposed method. One of the assumption is “A policy trained on a sufficient number of goals in some area of the goal-space will learn to interpolate to other goals within that area”. This seems to mean that the area is convex. It might be better if some quantitative analysis can be provided to illustrate geometry of goal space (given complex motor coordination) that is feasible for the proposed method.

4. It is difficult to understand the plots in Figure 4 without more details. Do you assume for every episode, the agent starts from the same state?

5. For the plots in Figure 2, is there any explanation for the large variance for Goal GAN? Given that the state space is continuous, 10 runs seems not enough.

6. According to the experimental details, three rollouts are performed to estimate the empirical return. It there any justification why three rollouts are enough?

7. Minor comments
Achieve tasks -> achieve goals or accomplish/solve tasks
A variation of to -> variation of
Allows a policy to quickly learn to reach …-> allow an agent to be quickly learn a policy to reach…
…the difficulty of the generated goals -> … the difficulty of reaching

---

> ### Author Response · Authors · 2017-12-27
> **Solving tasks with sparse rewards without modifying the reward function by automatically generating a curriculum over tasks.**
>
> We thank the reviewer for the thorough analysis and insightful comments. In the following we answer one by one the questions, and we detail the clarifications made in the paper wherever needed.
>
> 1. Our proposed method is able to solve tasks with sparse rewards without modifying the reward function by automatically generating a curriculum over tasks. As our Problem Definition (Sec. 3) states in the Overall Objective (Sec. 3.2), we are seeking a policy $\pi^*(\cdot | s_t, g)$ that can succeed at many goals $g$, each goal corresponding to a different task with its own sparse reward $r^g(s_t, a_t, s_{t+1})$. But, although all tasks have a sparse reward, they are not all of the same difficulty! In particular, reaching a goal nearby the starting position is very easy, and can be performed even by the randomly initialized policy. Then, once the policy has learned to  reach the nearby goals (in our navigation settings, it implies having learned some basic locomotor skills), it can bootstrap this acquired knowledge to attempt more complex (further away) goals. As explained in our Goal Labeling (Sec 4.1), our method strives to sample goals always of  “intermediate difficulty” $g: R_{\min} \leq R^g(\pi_i) \leq R_{\max}$. This means that our method will always be sampling goals such that training on them is efficient (i.e. our policy is able to receive a sufficient amount of reward such that it can improve its performance), despite their sparse reward structure. If no curriculum is applied, a prohibitively long time-horizon would be needed for the policy to learn to reach  the far away goals. Furthermore, many goals are actually infeasible, and no matter the time-horizon they always receive a reward of 0. Our method minimizes wasting rollouts trying to reach such goals because they do not satisfy our condition $R_{\min} \leq R^g(\pi_i)$.
>
> 2. The hyperparameters R_min and R_max have a very clear probabilistic interpretation given in Sec. 4.1, based on  analyzing Eq. (2). R_min is the minimum success probability required to start training on a particular goal. R_max is the maximum success probability above which we prefer to concentrate training on new goals. In practice, as explained in Sec. 4.3 and Appendix C, we estimate $R^g(\pi_i)$ with the rollouts collected by our RL algorithm. Therefore, each estimation is an average over two to five binary rewards (whether the rollout succeeded or not), meaning that the lowest numbers it can get are 0 or ⅕ and the highest are ⅘ or 1. In all our experiments we used R_min = 0.1 and R_max = 0.9, but given the above analysis any $R_min \in ]0, 0.2[$ and $R_max \in ]0.8, 1[$ would have yield exactly the same result. We have not experimented with values outside this range because it might not be of practical interest to not train on goals that are already achieved more than 20% of the time or have a policy succeeding less than 80% of the time on the goals it is given.
>
> 3. Our assumptions do not imply convexity of the goal space. For example, we do provide quantitative analysis for the Ant-Maze environment, where we report an efficient learning of our method despite the geometry of the feasible goal space being U-shaped, as seen in Fig. 4 (we have updated the legend to more clearly identify the feasible goal space). Rather, the interpolation statement refers to the smoothness of the goal space with respect to the policy, i.e. the policy for reaching a specific goal that has not been sampled during training can be inferred from sampling a sufficient number nearby goals in the continuous goal space.  The extrapolation statement should be understood along the lines of the explanation given in our point 1. of this rebuttal: “once the training policy is able to reach the nearby goals ... it can bootstrap this acquired knowledge to attempt more complex (further away) goals”. This is a very reasonable assumption in many learning systems, robotics in particular.

---

> > ### Author Response · Authors · 2017-12-27
> > **continued**
> >
> > 4. Indeed the agent starts from the same state at every rollout. Only the goal (and hence the reward) changes between rollouts. We have updated Fig. 4 to clearly mark the projection of the initial state onto the depicted x-y plane representing the Center of Mass positions. We hope this clarifies the plots.
> >
> > 5. We don’t think any of the methods presented have a significantly larger variance than the others. We agree that averaging over more than 10 random seeds would be desirable, although given time and compute constraints we couldn’t run more. Actually, 10 random seeds is considerably above standard in this field (most RL publications use 3 or 5 random seeds).
> >
> > 6. We apologize for a typo in Appendix B1-B2, where we stated that “For each goal, we estimate the empirical return with three rollouts”. This is only true for the ablation experiment  called “Goal GAN true label” shown in Appendix C, Fig. 6. For the “Goal GAN (ours)” method presented throughout the paper we do not sample more rollouts to label the goals; instead we reuse rollouts collected during the TRPO iterations. This means that the goals are labeled with a number of rollouts ranging from two to five (based on the number of times this goal was sampled during RL training). We have run an experiment of  sampling 10 additional rollouts to label every goal, and we observe that the performance does not differ significantly from the one already reported with three rollouts.
> >
> > Thank a lot for your additional comments, we have corrected all these typos in the paper.
> >
> > This review has been very helpful to improve the clarity of exposition. Please, let us know if any point is still unclear and we will very gladly extend our explanations.

---

### Decision · Program_Chairs · 2018-01-29
**ICLR 2018 Conference Acceptance Decision**

**Decision:**

Reject

**Comment:**

In principle, the idea behind the submission is sound: use a generative model (GANs in this case) to learn to generate desirable "goals" (subsets of the state space) and use that instead of uniform sampling for goals. Overall I tend to agree with Reviewer 3 in that the current set of results is not convincing in terms of it being able to generate goals in a high-dimensional state space, which seems to be be whole raison d'etre of GANs in this proposed method. The coverage experiment in Figure 5 seems like a good *illustration* of the method, but for this work to be convincing, I think we would need a more diverse set of experiments  (a la Figure 2) showing how this method performs on complicated tasks.

I encourage the authors to sharpen the definitions, as suggested by reviewers, and, if possible, provide experiments where the Assumptions being made in Section 3.3 are *violated* somehow (to actually test how the method fails in those cases).